# Operation of a Solid Oxide Fuel Cell Based Power System with Ammonia as a Fuel: Experimental Test and System Design

**Linda Barelli**, **Gianni Bidini and Giovanni Cinti** *

Department of Engineering, University of Perugia, Via G. Duranti 93, 06125 Perugia, Italy;
linda.barelli@unipg.it (L.B.); gianni.bidini@unipg.it (G.B.)
* Correspondence: giovanni.cinti@unipg.it

**Abstract:** Ammonia has strong potentialities as sustainable fuel for energy applications. $NH_3$ is carbon free and can be synthetized from renewable energy sources (RES). In Solid Oxide Fuel Cells, $NH_3$ reacts electrochemically thereby avoiding the production of typical combustion pollutants such as NOx. In this study, an ammonia-fueled solid oxide fuel cells (SOFC) system design is proposed and a thermodynamic model is developed to evaluate its performance. A SOFC short stack was operated with $NH_3$ in a wide range of conditions. Experimental results are implemented in the thermodynamic model. Electrical efficiency of 52.1% based on ammonia Lower Heating Value is calculated at a net power density of 0.36 W $cm_{FC}^{-2}$. The operating conditions of the after burner and of the ammonia decomposition reactor are studied by varying the values of specific parameters. The levelized cost of energy of 0.221 \$ $kWh^{-1}$ was evaluated, as introduced by the International Energy Agency, for a system that operates at nominal conditions and at a reference power output of 100 kW. This supports the feasibility of ammonia-fueled SOFC systems with reference to the carbon free energy market, specifically considering the potential development of green ammonia production.

**Keywords:** ammonia; SOFC; system; model; stack test

## 1. Introduction

The development of renewable energies, in particular wind and solar, requires the integration of energy storage solutions. Ammonia is a chemical used as a fertilizer but can be considered also as a fuel. Recently ammonia is recently presented as a potential energy storage solution [1,2]. Presently, ammonia is produced from natural gas but a pathway for the production of green ammonia can be designed based on hydrogen from electrolysis or biogas [3,4]. Table 1 compares ammonia with the most common fuels. Ammonia is characterized by a volumetric energy density significantly higher than compressed natural gas at 250 bar, with gravimetric energy density more than halved with respect to fossil fuels but greater than methanol.

**Table 1.** Energy densities of ammonia and other common fuels [5].

| | Liquid Ammonia | Pressurized Hydrogen (@350 bar) | Methanol | Natural Gas (@250 bar) | Gasoline | Diesel |
|---|---|---|---|---|---|---|
| Volumetric energy density – GJ $m^{-3}$ | 11.38 | 11.73 | 19.8 | 9.8 | 30.6 | 37.2 |
| Gravimetric energy density – MJ $kg^{-1}$ | 18.65 | 120.24 | 15.6 | 50 | 43.6 | 44.8 |

For the conversion of ammonia to power production, three main energy technologies are reported in the literature: internal combustion engines [6–9], gas turbines [10–12] and fuel cells [2,13,14].

Among these, solid oxide fuel cells (SOFCs) are the most interesting ones and whose development is presently reaching commercialization, especially in the power range of 1–250 kW and system electric efficiency above 60% based on natural gas low heating value (LHV). SOFCs operate at high temperature, 650–800 °C, reducing the cost of materials compared to low temperature fuel cells and producing heat that can be used for cogeneration application or recovered at system level to produce hydrogen through catalytic processes. This is the case of ammonia systems, where the heat can supply thermal energy to decompose ammonia. Moreover, the materials used as anodes in SOFCs are efficient catalyst for ammonia decomposition. Since both high temperature heat and catalytic material are already available in a SOFC system, it is possible to consider two types of system design: (i) ammonia can be decomposed in a specific reactor and the product gases, a mixture of hydrogen and nitrogen, are fed to the SOFC or (ii) ammonia can be directly fed to the SOFC stack. The latter is extremely interesting since the internal ammonia decomposition reaction absorbs the produced heat mitigating the need for the cooling of the SOFC power unit.

This study presents an innovative ammonia-fed SOFC system based on experimental results performed on a SOFC short stack. Literature reports studies relative to both experimental test and of power system designs of ammonia coupled with SOFC technology and study.

The experimental operation of SOFC with ammonia as a fuel is well assessed in the literature especially at the single cell level. While the first studies were dedicated to SOFCs with proton-conducting electrolyte [15–17], the so-called SOFC-H, recent studies are more focused on ion-conduction electrolyte, SOFC-O [18–20], due to higher level of maturity of the latter technology. Moving to the stack size, very little is reported in the literature in terms of experimental results. In [20], a stack operating with both ammonia and hydrogen was tested. A preliminary test was performed before testing the stack on two sealing materials and the $SiO_2$-$Al_2O_3$-$R_2O$-RO was selected as the most stable one. They found that the ammonia-fueled stack delivered 255 W at 56 A, reaching 53% efficiency based on the ammonia Lower Heating Value (LHV), which is almost the same performance as the one fueled with hydrogen. A second test was repeated on a stack consisted of 10 anode-supported planar cells each one with an active area of 95 cm$^2$ [21]. The stack was fed with pure ammonia and with two different decomposed ammonia mixtures, one produced with a cracking reactor and one with an auto-thermal reactor. Polarization curves show that decomposed ammonia has higher performance than pure ammonia at high current densities. Such behavior is explained by the steam produced by the electrochemical reaction that inhibits the decomposition of ammonia in the anode chamber. Pure $NH_3$ test achieved 232 W at 36 A with a total LHV efficiency of 36.3%. Low values of efficiency are mainly caused by low fuel use factor (Uf) used in the test (62% at 36 A). An endurance test of 1000 h was performed both with pure hydrogen and ammonia showing same degradation rate. No NOx was measured in the exhaust gases. In [22] a four cells stack was tested comparing pure hydrogen, pure ammonia, and decomposed ammonia mixture at three operating temperatures: 800, 750 and 700 °C. Polarization curves show no significant difference between the compositions at all the three tested temperatures. Finally, in [23], a 10 cells SOFC stack based on Electrolyte Supported Cells (ESC) and with chromium-based interconnects (CFY), was tested under four anodic feeding mixtures to compare ammonia and hydrogen performance with and without steam flow addition. Results show the equivalence between hydrogen and ammonia for both pure and humidified compositions. In addition, a durability test of 1000 h at 80% use of fuel was performed. Results showed the same degradation of 1.1%/1000 h for both ammonia and hydrogen. Ex situ analysis showed that degradation is caused by nitridation process of both interconnect and Ni contact mesh.

The literature also reports studies relative to NH$_3$-SOFC system analysis. Design studies here reported refer to different types of SOFC technologies and of power plants. This short review focuses mainly on the integration solutions presented in terms of balance of plant design. In [24] two different schemes are presented, each one integrates two heat exchangers, for both air and fuel pre-heat, and the after burner. The two designs differ from what concerns the air management strategy: required air can be fed directly to the stack, first design, or separated in two streams, bringing to a two-stacks

strategy, second design. In the second design the separated amount of air is mixed to first stack cathodic off-gases and fed to the cathode of the second stack. This design allows both to reduce the temperature of the mixture and to increase oxygen concentration of the second stack cathodic inlet. As a result, the cathodic heat exchanger size is halved. In [25] a portable system is studied. The after burner supplies the heat to a single heat exchanger that increases air temperature to stack inlet and, at the same time, supplies heat for the decomposition of ammonia. Water is also evaporated in the heat exchanger and added to the anode inlet gas mix. The model calculates up to 41.1% of efficiency for a fuel use of 0.8. A similar design is presented in [22] but implementing a three heat exchangers solution. Two separate heat exchangers operating at high temperature recover heat separately for air and fuel. The two separated gas flows are mixed at lower temperature in an after burner that completes the oxidation of the fuel and provides hot exhausts to feed a single heat exchanger. Such a component is designed for low temperature pre-heating of both anodic and cathodic gas flows. The study shows the advantages in terms of air flow reduction due to internal ammonia decomposition reaction. A system efficiency of 50% is calculated. In [26] a combined heat and power (CHP) system based on a SOFC-H and fed with ammonia is considered for vehicular applications. The hot gases from SOFC exhausts are mixed and split in two different gas flows to preheat ammonia and air inlet flows. The study focuses on energy and exergy analysis when varying operating parameters such as fuel use, current density, and stack temperature. The maximum efficiency of 48% is calculated when the SOFC-H operates at low current density.

Recently, a new design was proposed for a system operating with ammonia and SOFC to produce heat, hydrogen, and power [27]. Two different concepts are presented whether the SOFC electricity production is designed only to sustain electricity consumption of the system or to supply electricity externally. In both cases, the system operates at low fuel use since hydrogen is one of the products of the system. In the first concept, inlet gases are preheated, recovering heat from the catalytic burner and from an external heat source. The second concept has a two-stacks strategy and part of the anode off-gases delivered by the first stack feeds the second SOFC unit. Heat balance is guaranteed by the external heat source and by heat recovery from the catalytic burner off-gases to preheat both cathodic inlet flows and part of anodic inlet. Trigeneration efficiency is 81% for the first concept and 71% for the second one.

In general, the design optimization of SOFC systems fed by ammonia is only partially investigated in the literature. This study presents an innovative system design based on experimental results on a short stack. First, an experimental test was performed on a SOFC short stack operating with ammonia in a wide range of parameters variation. Secondly, test results were used to define the correlation between operating parameters and performance. The correlation was implemented in the system design, modeled with a calculation sheet. Subsequently, nominal conditions are identified and main parameters are varied to study the effect on the equilibrium. Finally, a preliminary economic feasibility study was performed using the levelized cost of energy (LCOE) parameter as introduced by the International Energy Agency in [28].

## 2. Methods

The methodology followed includes three main phases. First, the experimental activity was performed to provide experimental data, subsequently implemented in the modeling phase; accordingly, Section 2.1 describes the experimental methodology in terms of short stack description, test rig and test campaign. Second, the ammonia-fueled SOFC system was modeled implementing the methods and system features (e.g., system lay-out, main components and model equations) detailed in Section 2.2. Finally, the simulation outcomes were implemented in the LCOE evaluation procedure, developed according to the methodology presented in Section 2.3. A schematic of the methodological workflow is reported in Figure 1.

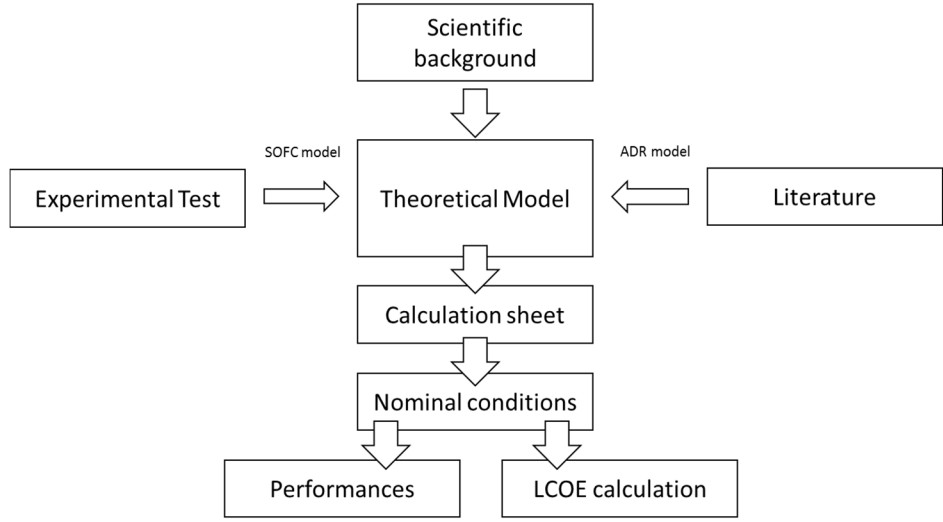

**Figure 1.** Schematic of the methodological workflow.

### 2.1. Experimental Description

To evaluate the performance of the ammonia-fed SOFC, an experimental campaign was designed and implemented on a SOFC short stack. The experimental study was performed on a six cells short stack supplied by SOLIDPower (SOLIDpower, Mezzolombardo, Italia). The stack is equipped with anode-supported planar cells. Details on stack design are reported in Table 2. Figure 2 is a picture of the test rig with the stack.

**Table 2.** SOFC stack design details.

| Parameter | Data |
|---|---|
| Number of cells – $n$ | 6 |
| Nominal Power | 100 W |
| Type | Anode-Supported Cell (ASC) – Planar |
| Dimension | active area ($A_{CELL}$) 80 $cm_{FC}^2$ |
| Anode | Ni/8YSZ 240 $\pm$ 20 $\mu$m |
| Electrolyte | 8YSZ 8 $\pm$ 2 $\mu$m |
| Bilayer cathode | GDC + LSCF 50 $\pm$ 10 $\mu$m |

The stack was operated on a laboratory test rig that allows controlling the stack operating temperature, gas flows, and operating current density. The test rig temperature and the cells voltage are measured with specific sensors placed in the stack and in the anodic and cathodic gas pipes, both inlet and outlet. Oxidant (air) and fuel (hydrogen, nitrogen, ammonia) flows are pre-heated inside the furnace, before entering the short stack. A detailed description of the test rig is reported in [29]. The aim of this study is to operate the stack at high efficiency and high power density. High efficiencies are achieved at high fuel use factor, which is the ration between the amount of hydrogen reacting electrochemically and the amount of hydrogen introduced in the stack (Equation (1)).

$$U_f = \frac{I * n}{2 * n_{H2EQ} * F} \tag{1}$$

where I is the operating current, n is the number of cells, F is Faraday constant and $n_{H2EQ}$ is the molar flow of equivalent hydrogen. In the case of pure ammonia, $n_{H2EQ}$ is equivalent to the amount of hydrogen obtained by the total decomposition of ammonia (reaction 2).

$$NH_3 \rightarrow \frac{1}{2}N_2 + \frac{3}{2}H_2 \tag{2}$$

Same approach can be used to calculate air flow fixing the oxidant use factor (Uox), as expressed by Equation (3).

$$U_{ox} = \frac{I \times x}{4 \times 0.21 \times n_{air} \times F} \quad (3)$$

were $n_{air}$ is the molar flow of air and 0.21 is the concentration of oxygen in the air. The experimental campaign was designed to evaluate stack performance by varying the following parameters: (i) current density, (ii) fuel use, (iii) ammonia decomposition rate (XNH3). XNH3 is useful to evaluate the performance of the stack when the decomposition of ammonia is partially or totally performed in an external decomposition reactor. Thus, it considers the amount of ammonia decomposition that occurs before entering the stack based on reaction (2), according to Equation (4) where $m_{NH3in}$ is the total inlet ammonia molar flow rate, while $m_{NH3out}$ is the ammonia molar flow rate at reactor outlet.

$$X_{NH3} = 1 - \frac{m_{NH3out}}{m_{NH3in}} \quad (4)$$

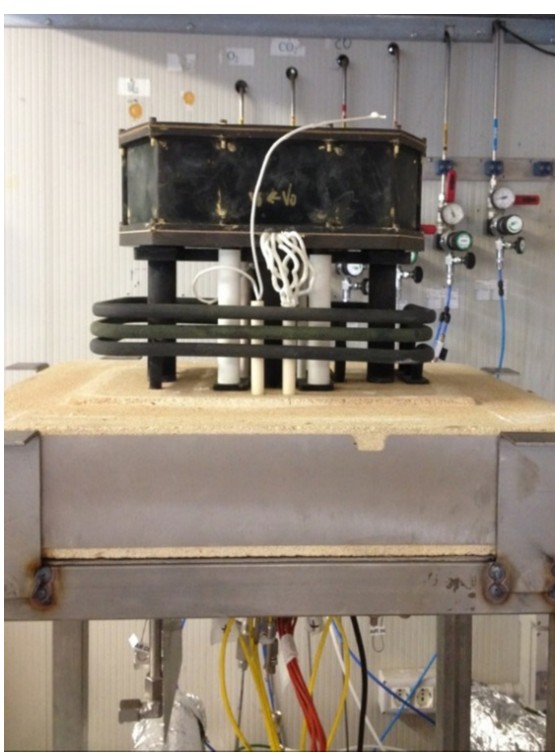

**Figure 2.** Photo of the short stack set-up.

Once the ammonia decomposition rate is defined, it is easily possible to calculate molar gas flow of ammonia, hydrogen and nitrogen entering the stack. These gas flows were fed to the short stack simulating the gas mix entering the unit when integrated with an external decomposition reactor. Table 3 reports constant and variable parameters used to define the test campaign.

**Table 3.** Test condition investigated in the experimental campaign.

| Parameter | Symbol | Unit | Values |
|---|---|---|---|
| Current density | J | mA cm$_{FC}^{-2}$ | 200–300–500 |
| Use of fuel | Uf | - | 0.6–0.7–0.8 |
| Ammonia decomposition | XNH3 | - | 0–0.5–1 |
| Temperature | TSOFC | °C | 750 |
| Use of oxygen | Uox | - | 0.2 |

During the experimental tests Uf was studied at three different values: 0.6, 0.7 and 0.8, while Uox was kept at 0.2. Current density was varied at values 200, 350 and 500 mA cm$_{FC}^{-2}$. Furnace temperature was kept constant at 750 °C, selected as the state of the art for the technology. The value selected for XNH3 are 0%, i.e., no external decomposition and pure ammonia in the stack, 50% and 100% corresponding to partial and total external decomposition, respectively. Based on the parameters in Table 2, it is possible to calculate gas flow rates for each operating point. For each test condition, the performance was evaluated in term of cell voltages after the stabilization time of 30 min.

## 2.2. Model Description

The system model was implemented on a Microsoft Excel© (Microsoft, Washington, DC, USA) calculation sheet using thermodynamic libraries taken from literature [30] and on the basis of the experimental activity outcomes, for what concerns the correlation between voltage and the main operating parameters such as current density, fuel use factor and ammonia decomposition rate. The model is zero dimensional and calculates the energy equilibrium of each component. The system scheme is reported in Figure 3. The proposed design moves from a standard state of the art solution of SOFC systems fed with natural gas. The fuel is decomposed in an external reactor and heat is recovered from the after burner off-gases (stream 10, downstream their mixing with bypassed air) to feed the reactor and to preheat air at high and low temperature.

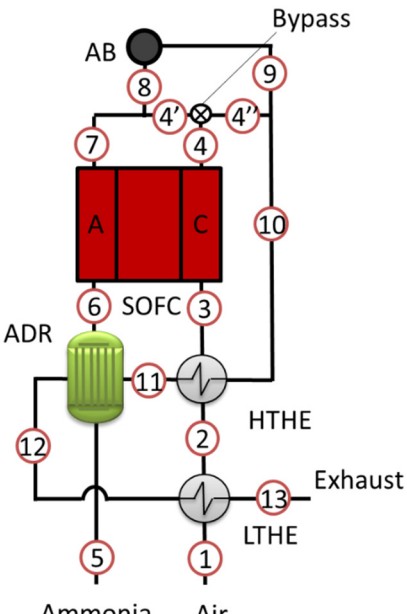

**Figure 3.** Scheme of the SOFC-NH$_3$ system.

The system is fed with pure ammonia and air and produces electrical power from the SOFC unit. The system is made of six components:

- SOFC: Solid Oxide Fuel Cell
- ADR: Ammonia Decomposition Reactor
- AB: After Burner
- HTHE: High Temperature Heat Exchanger
- LTHE: Low Temperature Heat Exchanger
- Bypass: Cathodic bypass

In addition, not reported in the scheme, the system requires an air blower and an inverter.

The SOFC unit is modeled as a reactor operating at constant temperature of 750 °C. Both anodic and cathodic gas flows enter the SOFC at 700 °C and exit at 800 °C. The gas composition is calculated

based on design parameters, i.e., current density and fuel use. The model considers the complete decomposition of ammonia in the anode according to reaction (2). This assumption is coherent with the local temperature and the presence of nickel as reaction catalyst. SOFC energy balance is calculated as follows (Equation (5)):

$$P_{SOFC} + \Delta h_{anode} + \Delta h_{cathode} - \dot{Q}_{SOFC} = 0 \tag{5}$$

where $P_{SOFC}$ is the SOFC electric power, $\Delta h_{anode}$ and $\Delta h_{cathode}$ are enthalpy flow differences between outlet and inlet of anodic and cathodic gases respectively. $\dot{Q}_{SOFC}$ represents the heat flow losses calculated as percentage, $HL_{SOFC}$ in Equation (6), of the total enthalpy flow differences both anodic and cathodic:

$$\dot{Q}_{SOFC} = HL_{SOFC} \times (\Delta h_{anode} + \Delta h_{cathode}) \tag{6}$$

$P_{SOFC}$ is calculated by multiplying current density and voltage, the latter is derived from experimental activity and deeply described in the following. The stack equilibrium model allows to calculate, as main output, the inlet cathodic air flow rate and, consequently, the use of oxygen.

The Ammonia Decomposition Reactor (ADR) is modeled as an adiabatic chemical reactor integrated with a heat exchanger. The required heat of reaction is provided by system hot gases (pipe 11 in the scheme). The energy balance of the ADR is calculated as reported in Equation (7):

$$\Delta h_{NH3} - \eta_{HE} \Delta h_{hot} = 0 \tag{7}$$

where $\Delta h_{NH3}$ is the enthalpy flow difference between outlet and inlet of the ammonia stream, $\Delta h_{hot}$ is the enthalpy flow difference between outlet and inlet of the hot gases and $\eta_{HE}$ is the efficiency of the heat exchanger, equal to 0.9. The same efficiency value is considered for the heat exchangers (HTHE, LTHE) implemented in the system. The decomposition rate, $X_{NH3}$, is calculated as function of reactor temperature, $T_{ADR}$, based on the literature [21]. Values considered where taken from the $Ni/Y_2O_3$ case catalyst. The ADR temperature varies from 450 °C to 650 °C. The reaction does not take place below the minimum temperature and, for those values, ammonia is not decomposed. Whereas above the higher value, 650 °C, ammonia is completely decomposed. The decomposition rate, as expressed by reaction (4), is obtained with a numeric approach as regression of literature experimental data [23] as reported in Equation (8) where $T_{ADR}$ must be expressed in °C:

$$X_{NH3} = 12.72 - 7.46 \times 10^{-2} \times T_{ADR} + 1.4 \times 10^{-4} \times T_{ADR}^2 + 8.4 \times 10^{-8} \times T_{ADR}^3 \tag{8}$$

To distribute the heat between HTHE and LTHE, and calculate equilibrium parameters, it is necessary to set one temperature parameter that controls and optimize energy recovery in the system. The control parameter is the ADR approach, $ADR_A$ that is the temperature difference between hot temperature inlet, pipe 11, and fuel ADR outlet temperature, pipe 6 in the scheme, corresponding to cathodic inlet temperature of 700 °C.

The after burner is modeled as an adiabatic reactor where the combustion reaction is completed. Depending on the composition of inlet flow, pipe 8 in Figure 3, the limiting reagent of the combustion can be either oxygen or hydrogen. Please note that the bypass allows controlling the amount of oxidant flow rate to the afterburner and, consequently, the O2/H2 ration called $\lambda_{AB}$. The control parameter is the bypass open rate, BOR, defined as the ration between the flow rate flowing to the afterburner and the total cathodic off-gases. This strategy permits to operate the after burner with different rate of O2/H2, including the no combustion option ($\lambda_{AB} = 0$), with the total air flow rate bypassing the after burner. The air blower is designed based on air flow rate $\dot{N}_{air}$ and total pressure losses of cathodic pipes, $\Delta Pc$. Blower efficiency was estimated at 0.9 and pressure losses of 0.15 bar. Ammonia is usually stored in liquid form in compressed cylinders (c.a. 10 bar) and no active system is required to pressurize the anodic line. The inverter power conversion efficiency, $\eta_I$, was set to 0.95. Table 4 reports all inlet parameters used to design the system while Table 5 reports system constant parameters.

**Table 4.** Inputs and outputs parameters of the model.

| Inlet Parameter | Symbol | Unit |
|---|---|---|
| Use of fuel | Uf | - |
| Current density | J | A cm $_{FC}$ $^{-2}$ |
| ADR decomposition temperature | $T_{ADR}$ | °C |
| Stack heat losses | $HL_{SOFC}$ | % |
| Bypass Open Rate | BOR | % |
| ADR approach | $ADR_A$ | °C |

**Table 5.** Constant parameters in the model.

| Constant Parameters | Symbol | Unit | Value |
|---|---|---|---|
| Blower efficiency | $\eta_B$ | | 0.9 |
| Heat exchangers efficiency | $\eta_{HE}$ | | 0.9 |
| Cathodic losses | $\Delta Pc$ | bar | 0.15 |
| SOFC temperature | TSOFC | °C | 750 |
| Inverter efficiency | $\eta_I$ | | 0.95 |

*2.3. LCOE Model*

This study aims also at assessing the economic advantages of the proposed system. This part of the study was approached by evaluating the levelized cost of energy (LCOE). The LCOE allows calculating and comparing the convenience of different typologies of power plants. Basically, the LCOE relates the total cost of a power plant, including operational costs, and the total amount of energy produced. In the calculation sheet, in which the model was studied, a section for the levelized cost of energy (LCOE) calculation was implemented. The results from the thermodynamic study, in terms of components dimensions and energy produced, were used as inputs for the LCOE study. The calculation is based on Equation (9):

$$LCOE = \frac{\sum_c \left[\left(\sum_a \frac{OM_a}{(1+r)^a}\right)_c + I_c\right]}{\sum_a \frac{E_a}{(1+r)^a}} \tag{9}$$

where OMa represents operation and maintenance costs for the component c and the year a; r is the discount rate; $I_c$ is the investment cost of the component c; $E_a$ is the electrical energy generated in the year a. The cost calculation of each component was performed based on the literature [31–35]. Equations used in the model are reported in Table 5 with their respective reference sources. The cost, $I_n$, is reported for each component previously defined and reported in the first column of the table. It is important to highlight that these equations are a literature reference that cannot completely describe technologies in a development phase; moreover, the cost of each component is extremely variable. This comment can refer to the most innovative components, such as the SOFC stack, but possibly also to technologies that are state of the art, such as heat exchangers, inverters, and blowers. The explanation is that a new application can bring innovation also in the standard components that need to be implemented and customized for the specific design. In addition, the reported equations can hardly follow the size related costs. For example, the stack equation is linear with the area, while more detailed models show a decrease of specific cost with the size increase [34]. Finally, cost prediction is extremely correlated with the development of the market and a more detailed model can be implemented considering the number of units produced per year. Nevertheless, the equations reported in Table 6 are homogenous and coherent and allow comparing different power system designs based on the same SOFC technology. Since it is not easy to evaluate the issues related to the use of ammonia, also maintenance costs are difficult to be predicted. Nonetheless, degradation of materials should improve since a carbon free fuel, ammonia, is used. Moreover, the elimination of water line and of the steam methane reformer reactor contributes to the reduction of maintenance costs. On the other

hand, there is no experience about material degradation due to other causes such as nitridation of steel materials. For the present study, the main data were taken from literature based on a 100 kW SOFC system operating with natural gas. Maintenance costs used for the calculations are reported in 1 in the equation: $air_{in}$: AB inlet air flow, $AB_{dp}$: AB pressure losses, $T_{out}$: AB outlet exhaust temperature.

**Table 6.** Capital cost function of system components.

| System Component | Capital Cost Function ($) | Ref |
|---|---|---|
| SOFC stack | $I_{SOFC} = n \cdot A_{cell} \cdot (2.96 T_{SOFC} - 1907)$ | [31,33] |
| Auxiliary devices | $I_{SOFC,aux} = 0.1(I_{SOFC})$ | [31,33] |
| After Burner[1] | $I_{AB} = \left(\frac{46.08\ air_{in}}{0.995 - AB_{dp}}\right)[1 + \exp(0.018 \cdot T_{out} - 26.4)]$ | [31,33] |
| Blower | $I_B = 91.562\left(\frac{W_B}{455}\right)^{0.67}$ | [32] |
| Heat Exchangers | $I_{HE} = 130\left(\frac{A_{HE}}{0.093}\right)^{0.78}$ | [33,35] |
| Inverter | $I_I = 10^5\left(\frac{W_{SOFC}}{500}\right)^{0.7}$ | [31–33] |

[1] in the equation: $air_{in}$: AB inlet air flow, $AB_{dp}$: AB pressure losses, $T_{out}$: AB outlet exhaust temperature.

Table 7 together with the constant parameters included in the LCOE definition as expressed by Equation (9), specifically:

- discount rate, r;
- system availability (in percentage) defined as the amount of time, year-based, the system is on; this parameter is necessary to calculate real energy generated in the year (Ea) and the related fuel consumption;
- maintenance costs, as already commented;
- fuel cost.

**Table 7.** LCOE constant parameters.

| Parameter | Value | Ref |
|---|---|---|
| Discount rate | 8% | [32,36] |
| Lifetime of overall system | 20 years | [32] |
| System availability | 80% | [32] |
| Maintenance costs | 36752 € $y^{-1}$ | [32] |
| $NH_3$ cost | 272,5 € $kg^{-1}$ | [37] |

The area $A_{HE}$ of the heat exchangers is calculated based on the definition provided by Equation (10).

$$A_{HE} = \frac{Q_{HE}}{LMTD \times U} \tag{10}$$

where $A_{HE}$ it the heat exchanger area, $Q_{HE}$ is the exchanged thermal power, LMTD is the logarithmic mean temperature difference and U is the exchange coefficient fixed at 30 W $m^{-2}$ $K^{-1}$.

## 3. Results

The results session is divided between experimental results, model system study, and LCOE analysis. Experimental results were used to develop the stack model integrated into the overall system model. System model results were implemented in the LCOE analysis.

### 3.1. Experimental Results

The tests performed on the short stack, as defined in Table 3, cover 27 operating conditions. Tests scheduling was planned to reduce stack shock due to the change of operating conditions. Parameters were changed according to the hierarchy order: $X_{NH3}$, fuel use and current density. Temperature was kept constant at 750 °C. $X_{NH3}$ was changed in the order 0%, 50% and 100%, while fuel use changed for each $X_{NH3}$ value in the order 0.8, 0.7 and 0.6. Finally, current density was changed, where the test starts from higher current density, 500 mA $cm_{FC}^{-2}$, and higher fuel use of 0.8, considered to be the most stressing for the stack. Subsequently, current was reduced down to minimum current density and then the gas conditions were fixed for the following Uf condition, 0.7, and then current was raised back to the higher values. Finally, the current density was raised back at 500 mA $cm_{FC}^{-2}$ and new Uf condition, 0.6, was introduced and then current was reduced stepwise down to lower values. Figure 4 reports, as example, the part of the test at 750 °C and $X_{NH3}$ of 50%. The graph shows the data relative to the three fuel use factors and, for each Uf, to the three current density values. The graph reports the average value of the cells voltages and the cathodic temperature measured at the outlet. Regarding stack temperature, the design temperature was set to the furnace regulating system. This means that the measured stack temperature, as reported in the graph, is not fixed but varies, in a short range, with the operating conditions. Cathodic outlet temperature is, in general, considered to be the closest to stack average temperature. In Figure 4, it is also possible to note that the voltage is quickly stabilized, while temperature has longer response time and equilibrium is not always reached.

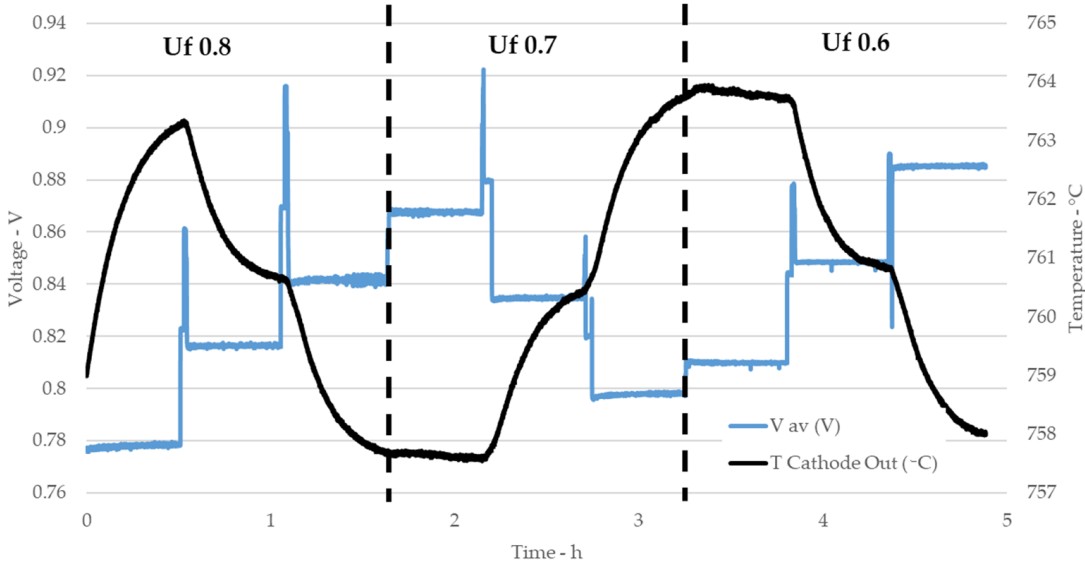

**Figure 4.** Example of stack test: average cell voltage and cathode temperature as function of time.

In the following analysis, the results reported refer to the average value of the six cells and calculated as average of 60 s of samples, taken at 1 Hz. To avoid the overlapping with following phase, the 60 s considered for the calculations are two minutes before the new test condition starts. Figure 5 reports the main results in terms of average cell voltage value as function of current density for fuel use of fuel 0.8 (a), 0.7 (b) and 0.6 (c). Each graph reports three curves, one for each $X_{NH3}$ value. As expected, for all curves, voltages decrease with increasing current and higher Uf values corresponds to lower voltages. Regarding $X_{NH3}$, there are no significant differences in terms of measured voltage between the three compositions. At higher current density, voltages values of decomposed ammonia are lower compared to ammonia containing mixtures. This trend is coherent for all Uf values and even though the difference is low, pure ammonia tests are slightly more performing than equivalent mixtures.

Stack performance in terms of efficiency and power density are reported in Figure 6a,b respectively. The curves report values for different Uf at the constant temperature of 750 °C and constant *XNH3 =* 0. Since voltages values of the other decomposition rates are extremely close each other, also power densities and efficiencies are similar and not reported. The efficiency of 56% at the higher power density of 38.6 W cm$_{FC}$$^{-2}$ was obtained with pure NH$_3$.

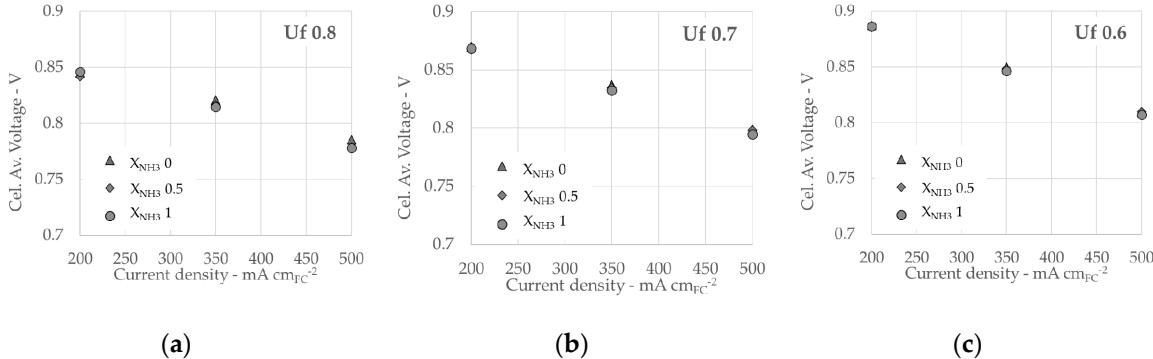

**(a)** **(b)** **(c)**

**Figure 5.** Stack test results at 750 °C for Uf 0.8 (**a**), Uf 0.7 (**b**) and Uf 0.6 (**c**).

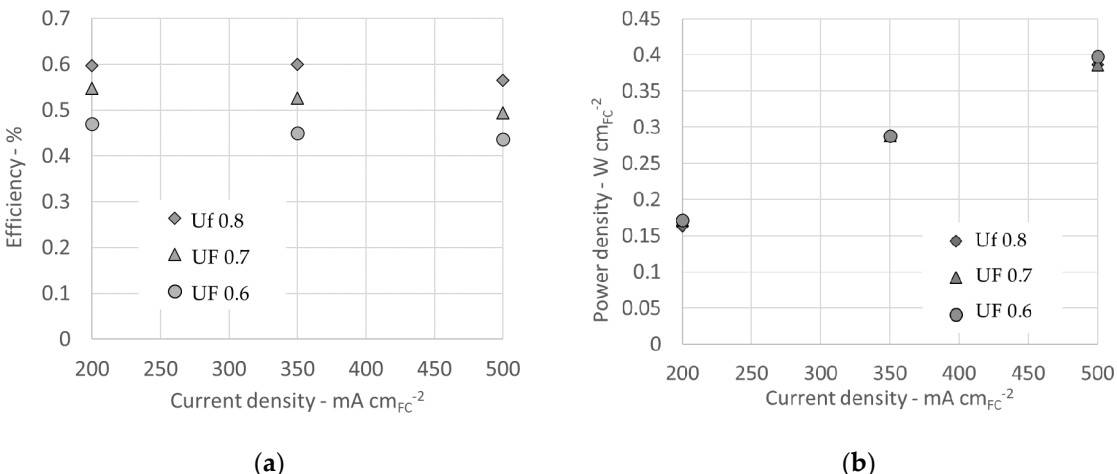

**(a)** **(b)**

**Figure 6.** Stack efficiency (**a**) and power density (**b**) for pure ammonia (XNH3 = 0) and temperature 750 °C.

Based on experimental results it was possible to define a function that relates voltage with the investigated parameters. Linear regression was always considered for each parameter. Due to the inconsistency of temperature, regression was performed separately for each temperature. At 750 °C the regression was calculated as expressed by Equation (11).

$$V(Uf, J, X_{NH3}) = -0.1686 \, Uf - 2.3638 \cdot 10^{-4} J - 3.2182 \cdot 10^{-5} X_{NH3} + 1.0342 \tag{11}$$

Please note that J is introduced in mA cm$_{FC}$$^{-2}$ and X$_{NH3}$ assumes values in the range between 0 and 1. As expected, the correlation is negative for both Uf and J since the voltage decreases when both parameters increase. The influence of the ammonia decomposition parameter is negligible.

Emissions from the stack were sampled and analyzed. Table 8 reports the values measured at 750 °C and pure ammonia (X$_{NH3}$ = 0), where the table reports the operating conditions and the NH$_3$ concentration measurements are given. Moreover, further parameters are reported to correlate the emissions to additional operating conditions. The specific emission evaluation is provided per NH$_3$ inlet flow rate and total current, i.e.,

- $S_{NH3}$ defined as the ration between $NH_3$ emissions concentration and ammonia inlet flow rate;
- $R_{NH3}$ determined as the ration between emissions and total current.

Even though it is not possible to give a complete description of the relation between emissions and operating conditions from the current measurements, a strong correlation between ammonia flows and emission emerges. Low values of ammonia inlet flow rates, such as the ones at 16 A, corresponds to much lower emissions compared to the flow rates related to the tests performed at 40 A. This tendency is described by parameter $S_{NH3}$, with values close to one for small flows (i.e., at 16 A) and a coherent increase at higher values. Looking to values at 40 A it is clear, as described also by the $R_{NH3}$ parameter that different emission levels correspond to the same current density value. The results show that unlike what is reported in [21], the influence of ammonia inlet flow on the $NH_3$ emissions is greater than of Uf.

**Table 8.** Off-gas measurements of $NH_3$ emissions for pure ammonia tests.

| Uf | Uox | $m_{NH3}$ Nl h$^{-1}$ | $m_{Air}$ Nl h$^{-1}$ | I A | J mA cm$_{FC}$ $^{-2}$ | $NH_3$ ppm | $S_{NH3}$ ppm Nl h$^{-1}$ | $R_{NH3}$ ppm A$^{-1}$ |
|------|------|------|---------|----|-----|------|------|-------|
| 0.80 | 0.20 | 83.63 | 1194.71 | 40 | 500 | 260 | 3.11 | 6.50 |
| 0.80 | 0.20 | 33.45 | 477.89 | 16 | 200 | 40 | 1.20 | 2.50 |
| 0.60 | 0.20 | 111.51 | 1194.71 | 40 | 500 | 1000 | 8.97 | 25.00 |
| 0.60 | 0.20 | 44.60 | 477.89 | 16 | 200 | 45 | 1.01 | 2.81 |

### 3.2. Performance Analysis of the SOFC-NH$_3$ System

The experimental results of the stack were implemented in the theoretical model, as defined in Equation (10). The system was studied and nominal conditions, as reported in Table 9, were identified. Electrochemical operating conditions of the SOFC stack were fixed with the selection of fuel use of 0.8 and current density of 0.5 A cm$_{FC}$$^{-2}$. These values represent a standard trade-off between power density that requires high values of current density, and efficiency that requires small currents. ADR temperature was set to 350 °C, meaning that no decomposition is considered in the ADR and the chemical reaction takes place totally inside the stack. The BOR was set to one, meaning that all the air coming from the cathode outlet goes into the after burner. The ADR approach was fixed to 20 °C.

**Table 9.** Inputs values of system nominal conditions.

| Parameter | Unit | Value |
|-----------|------|-------|
| Uf | | 0.8 |
| J | A cm$_{FC}$ $^{-2}$ | 0.5 |
| $T_{ADR}$ | °C | 350 |
| $HL_{SOFC}$ | - | 0.05 |
| BOR | - | 1 |
| $ADR_A$ | °C | 20 |

Table 10 reports main results of the system operation under nominal condition. The stack energy balance requires an oxygen use of 0.17, corresponding to a specific air flow rate of 2.93 Nl h$^{-1}$ cm$_{FC}$$^{-2}$. The stack operates at cell voltage of 0.78 V producing a power density of 0.36 W cm$_{FC}$$^{-2}$. System gross efficiency is 54.2% while net efficiency is 52.1%. Since all ammonia decomposition takes place inside the stack, $X_{NH3}$ is zero. The lambda of the after burner is 9.78, as a consequence of flowing all cathodic exhausts into the after burner.

**Table 10.** Outputs parameters values of system nominal conditions.

| Parameter | Symbol | Unit | Value |
|---|---|---|---|
| Use of oxygen | $U_{ox}$ | - | 0.17 |
| Cell voltage | $V$ | V | 0.78 |
| Power density | $P_{SOFC}$ | $W\ cm_{FC}^{-2}$ | 0.36 |
| Net efficiency | $\eta_{net}$ | - | 52.1% |
| Gross efficiency | $\eta_{gross}$ | - | 54.2% |
| ADR reaction rate | $X_{NH3}$ | - | 0 |
| Off-gases temperature | $T_{13}$ | °C | 192.60 |
| Heat Exchangers total area | $a_{THE}$ | $cm_{HE}^2\ cm_{FC}^{-2}$ | 1.67 |
| $NH_3$ flow rate | $m_{NH3}$ | $g\ h^{-1}\ cm_{FC}^{-2}$ | 0.13 |
| Air flow rate | $\dot{N}_{air}$ | $Nl\ h^{-1}\ cm_{FC}^{-2}$ | 2.93 |
| Lambda After Burner | $\lambda_{AB}$ | - | 9.78 |

Table 11 reports gas compositions for all pipes. Please note that hydrogen is completely oxidized in the after burner and off-gases contain only oxygen, nitrogen, and water (steam).

**Table 11.** Outputs parameters values of system in nominal conditions.

|  | Air (1,2,3) | $NH_3$ in (5) | $NH_3$ Dec. (6) | Anode Out (7) | Cathode Out (4,4′,4″) | AB Mix (8) | AB Out (9) | Off-Gases (9,10,11,12.13) |
|---|---|---|---|---|---|---|---|---|
| $H_2O$ | 0.0% | 0.0% | 0.0% | 60.0% | 0.0% | 6.6% | 8.4% | 8.4% |
| $N_2$ | 79.0% | 0.0% | 0.0% | 25.0% | 81.9% | 75.7% | 76.9% | 76.9% |
| $H_2$ | 0.0% | 0.0% | 0.0% | 15.0% | 0.0% | 1.6% | 0.0% | 0.0% |
| $NH_3$ | 0.0% | 100.0% | 100.0% | 0.0% | 0.0% | 0.0% | 0.0% | 0.0% |
| $O_2$ | 21.0% | 0.0% | 0.0% | 0.0% | 18.1% | 16.1% | 14.7% | 14.7% |

Table 12 reports parameters values of the heat exchangers, including ADR. For each component inlet and outlet temperature of both hot and cold gas flows are reported. The table reports also LMTD, specific heat exchanged in the heat exchanger, $q_{HE}$, U and specific area of the heat exchanger, $a_{HE}$, of each heat exchanger. Both values of $q_{HE}$ and $a_{HE}$ are specific per unit of fuel cell area. The total specific area of heat exchangers, as reported in Table 10, is 1.67 $cm_{HE}^2\ cm_{FC}^{-2}$.

**Table 12.** Data of the heat exchangers.

| Parameter | Unit | LTHE | HTHE | ADR |
|---|---|---|---|---|
| $T_{hot\ in}$ | °C | 662.30 | 919.87 | 720.00 |
| $T_{hot\ out}$ | °C | 192.60 | 720.00 | 662.30 |
| $T_{cold\ in}$ | °C | 20.00 | 496.94 | 20.00 |
| $T_{cold\ out}$ | °C | 496.94 | 700.00 | 700.00 |
| LMTD | °C | 168.96 | 221.46 | 179.37 |
| $q_{HE}$ | $W\ cm_{FC}^{-2}$ | 0.58 | 0.27 | 0.07 |
| U | $W\ m^{-2}\ K^{-1}$ | 30 | 30 | 30 |
| $a_{HE}$ | $cm_{HE}^2\ cm_{FC}^{-2}$ | 1.15 | 0.40 | 0.13 |

The system was studied also in the neighborhood of the nominal condition. Figure 7 reports results relative to BOR variation keeping constant all other input parameters. In detail Figure 7a shows the trend of $\lambda_{AB}$ and of the system off-gases temperature as function of BOR. The increase in BOR corresponds to the increase of oxidant flow into the afterburner. Off-gases temperatures show a significant initial increase and a plateau for BOR value of 0.05 corresponding to the stoichiometric $\lambda_{AB}$ value of 0.5. Below this value, the combustion of hydrogen is not completed and all system temperatures decrease, including the off-gases temperature that reaches a minimum value of 57.4 °C. This specific aspect is deeper described by the graph reported in Figure 7b where $a_{THE}$ and chemical losses are reported. Chemical losses are calculated as the ratio between energy content in the off-gases,

in terms of hydrogen lower heating value, and total inlet ammonia energy. Both curves show an initial decrease, for BOR values lower than 0.05, and a successive stabilization. If no air reaches the afterburner, hydrogen is not combusted and all heat exchangers operate at lower temperatures. Consequently, LMTD decreases for each heat exchanger component and $a_{THE}$ increases. Regarding the chemical losses, for BOR equal to zero, hydrogen in the stack exhausts will be vented in atmosphere. Such amount corresponds to chemical losses of 22.7% of total energy input. When the after burner reaches values bigger than stoichiometric, $a_{THE}$ remains constant and chemical losses are null, since no hydrogen is vented. This study shows that the system can operate without the afterburner. This is an interesting peculiarity of this system, derived by the use of ammonia as a fuel. Hydrogen is not a harmful gas for the environment and the emission is not limited by environmental regulations. On the one hand, it possible to eliminate the after burner and the potential emissions typical of all combustion reactions, including the hydrogen one. On the other hand, the lower values of heat exchangers temperature brings to an increase of total surface of the components. Since the BOR variation does not modify the electrochemical performance and ammonia consumption, the net efficiency remains constant at 52.1%.

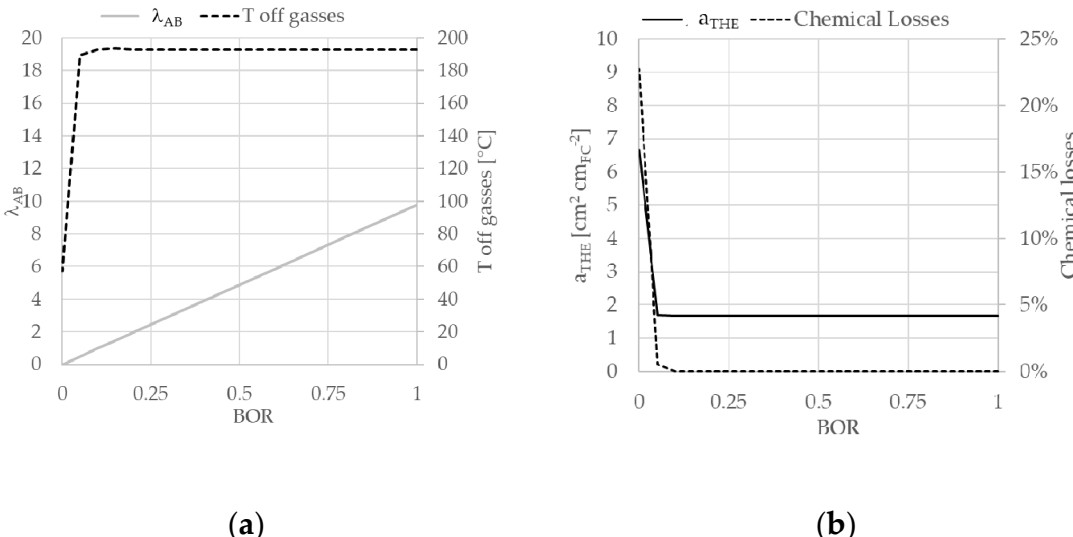

**(a)**　　　　　　　　　　　　　　　　　**(b)**

**Figure 7.** Graphs reporting results of BOR study results: $\lambda_{AB}$ and T off-gases (**a**) and $A_{THE}$ and chemical losses (**b**).

Figure 8 shows the study of $T_{ADR}$ variation, where Figure 8a reports the use of oxygen and net efficiency variation while Figure 8b reports off-gases temperature. The variation of $T_{ADR}$ from 350 °C to 650 °C moves the decomposition reaction from the stack to the ADR. The result is a decrease in Uox, since there is no cooling effect of the reaction inside the stack. Efficiency slightly decreases due to the effect of $X_{NH3}$ in Equation (10).

Based on the model results in nominal condition it was possible to calculate absolute values of the main parameters for a 100 kW system, shown in Table 13. The total area of SOFC is 28.04 m², while the total area of heat exchangers is 46.67 m². The table also reports absolute values of ammonia and air flow rates.

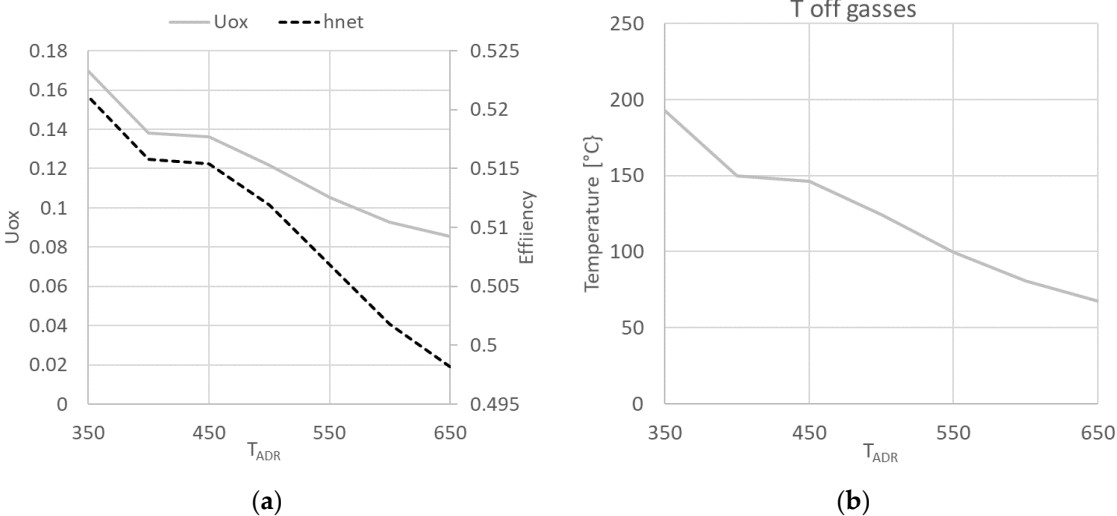

**Figure 8.** Variation of Uox and $\eta_{net}$ (**a**) and of off-gases temperature (**b**) as function of $T_{ADR}$.

**Table 13.** Absolute parameters values for a 100 kW system.

| Parameter | Unit | Value |
|---|---|---|
| System Power | kW | 100 |
| Total $A_{SOFC}$ | $m^2$ | 28.04 |
| Total $A_{THE}$ | $m^2$ | 46.67 |
| $NH_3$ flow rate | $g\,s^{-1}$ | 10.29 |
| Air flow rate | $Nl\,s^{-1}$ | 228.28 |

*3.3. LCOE Analysis of the SOFC-NH$_3$ System according to Different Designs*

LOCE study was dedicated to nominal condition and based on the simulation results presented in the previous Section 3.2. For the calculations, constant parameters were used as reported in Table 5. In addition, the study considers the availability of ammonia and no cost for its distribution and storage was considered. The LCOE, according to the procedure detailed in Section 2.3, results in 0.221 $ kWh$^{-1}$ for nominal design conditions. The capital cost of each component and relative quote for a 100 kW system is reported in Table 14. The most significant share of the cost fall in the stack and in the inverter. It is notable that low temperature heat exchange has a much higher cost that the high temperature one due to the higher exchange area required to complete heat transfer at lower temperature (see $a_{HE}$ values in Table 12).

**Table 14.** Component costs of a 100 kW system operating in nominal condition.

| Component | Value ($) | Quote |
|---|---|---|
| Stack | 31,444.41 | 34.4% |
| Stack AUX | 3144.44 | 3.4% |
| High Temperature Heat Exchanger | 5463.60 | 6.0% |
| Ammonia Decomposition Reactor | 2207.34 | 2.4% |
| Low Temperature Heat Exchanger | 12,408.59 | 13.6% |
| After Burner | 436.45 | 0.5% |
| Blower | 3915.19 | 4.3% |
| Inverter | 32,413.13 | 35.5% |
| TOT | 94,790.50 | - |

It is possible to compare the obtained LCOE value with the literature. In particular, in [32] a 100 kW SOFC system fed by natural gas reports a LCOE of 0.264 $ kWh$^{-1}$. The reference system is equivalent to the ammonia system here presented, with a slightly lower electrical efficiency of 51.7%. The value, as reported in [32], is calculated with reference to a Korean fuel cost of 16.61 $ GJ-LHV$^{-1}$. It is important to comment that the fuel cost plays an important role in the LCOE calculation. The value considered in the reference is in the range of household market. If we look to the European industrial market, the cost of natural gas decreases down to 7.88 $ GJ-LHV$^{-1}$ (1.11 €/$ exchange rate) [38]. In this sense the ammonia value of 16.37 $ GJ-LHV$^{-1}$ [37] is closer to household natural gas market. This explains why the LCOE obtained in the current work is lower than the one in the reference. Moving to the industrial market, this could be more meaningful due to the size of the system, the LCOE of natural gas system will drop down to 0.194 $ kWh$^{-1}$ since the fuel cost share is 50% out of the total. Results are coherent, considering that today ammonia in the market is produced from natural gas. As already explained in the introduction, the ammonia convenience has to be considered in the carbon free energy market and the potential development of green ammonia production.

## 4. Discussion and Conclusions

This study presents the design and modeling of an ammonia-fed SOFC system based on experimental campaign on a six cells SOFC short stack. An innovative design is presented, and the relative model was implemented to calculate thermodynamic parameters.

The experimental study of operating conditions of a SOFC short stack fed with ammonia was performed. The correlation with use of fuel, current density, and ammonia decomposition was studied at an operating temperature of 750 °C. The external ammonia decomposition has a minimum influence on performance, meaning that internal cracking of ammonia in the stack is a feasible solution. The stack achieves up to 60% efficiency at 750 °C. Measurements done on the ammonia emissions show NH$_3$ content in the range 40–250 ppm when moving to the highest current densities.

The system model allowed to calculate up to 52.1% of net efficiency in nominal condition. The parameter study showed how external decomposition of ammonia increases the size of the heat exchangers with no advantages in terms of efficiency. The most feasible strategy to variate system power is to rate the current density. Reduction of current density improves the efficiency of the system and increases oxygen use.

System LCOE of 0.221 $ kWh$^{-1}$ was calculated for a 100 kW system operating in nominal conditions. It is important to underline that the LCOE study is based on literature reference but the cost of components are strongly dependent on market development and technology innovations.

The study and the model developed in this study constitutes an important support for the design of an ammonia-fed SOFC energy system and provides indication for the sizing of the power unit.

**Author Contributions:** Conceptualization, G.C.; methodology, G.C. and L.B.; software, G.C.; validation, G.C.; formal analysis, G.C. and L.B.; investigation, G.C.; resources, G.B. and G.C.; data curation, G.C.; writing—original draft preparation, G.C.; writing—review and editing, G.C., G.B. and L.B.; visualization, G.C. and L.B.; supervision, G.C. and L.B.; project administration, G.C. and G.B.; funding acquisition, G.C. All authors have read and agreed to the published version of the manuscript.

**Funding:** The study was supported by the Fuel Cell and Hydrogen 2 Joint Undertaking under grant agreement No. 736,648 (NET-Tools project). The activities were also supported by ENVIU, PROTON VENTURES BV and C-JOB & PARTNERS BV, in the frame of a scientific collaboration.

**Acknowledgments:** The study was supported by the Fuel Cell and Hydrogen 2 Joint Undertaking under grant agreement No. 736,648 (NET-Tools project). The activities were also supported by ENVIU, PROTON VENTURES BV and C-JOB & PARTNERS BV, in the frame of a scientific collaboration.

**Conflicts of Interest:** The authors declare no conflict of interest.

## Abbreviations

The following abbreviations and symbols are used in this manuscript:

| Abbreviation | Description | Unit |
|---|---|---|
| AB | After Burner | |
| $A_{CELL}$ | SOFC cell active area | $cm_{FC}^2$ |
| ADR | Ammonia Decomposition Reactor | |
| $ADR_A$ | ADR approach | °C |
| $A_{HE}$-$a_{HE}$ | Area of the Heat Exchanger - specific per fuel cell area | $cm_{HE}^2$-$cm_{HE}^2$ $cm_{FC}^{-2}$ |
| $A_{SOFC}$ | Total SOFC active area | $cm_{FC}^2$ |
| ASR | Area Specific Resistance | $\Omega$ $cm_{FC}^2$ |
| BOR | Bypass Open Rate | |
| CHP | Combined Heat and Power | |
| $D_{NH3}$ | Ammonia decomposition | |
| HE | Heat Exchanger | |
| HTHE | High Temperature Heat Exchanger | |
| J | Current density | $mA$ $cm_{FC}^{-2}$ |
| LCOE | Levelized Cost Of Energy | $\$$ $kWh^{-1}$ |
| LMTD | Logarithmic Mean Temperature Difference | °C |
| LTHE | Low Temperature Heat Exchanger | |
| OM | Operation and Management | |
| $Q_{HE}$-$q_{HE}$ | Heat exchanged in the Heat Exchanger - specific per fuel cell area | W-W $cm_{FC}^{-2}$ |
| RES | Renewable Energy Sources | |
| SOFC | Solid Oxide Fuel Cell | |
| SOFC-H | Proton-conducting SOFC | |
| SOFC-O | Ion conducting SOFC | |
| $T_{ADR}$ | ADR decomposition temperature | °C |
| *TSOFC* | Stack Temperature | °C |
| U | Thermal exchange coefficient | $W$ $m^{-2}$ $K^{-1}$ |
| $U_f$ | Utilization of Fuel | |
| *Uox* | Utilization of oxygen | |
| $\eta_{net}$ | Net efficiency | |
| $\eta_{gross}$ | Gross efficiency | |

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
