# Peer review of "Operation of a Solid Oxide Fuel Cell Based Power System with Ammonia as a Fuel: Experimental Test and System Design"

_energies, doi:10.3390/en13236173_

Round 1

Reviewer 1 Report

The authors mentioned that ammonia could be produced using renewable energy sources. Please provide some references and methodologies in the paper.

What are the potential environmental impacts of using ammonia in SOFC such as in mass production, storage tank and materials, toxicity, etc.?

Please include a graph showing the data and the correlation (Eq. (8)) ?

In Table 13, low temperature heat exchanger cost a lot more than the other heat exchanger? Any reasons?

What is the most economical value of LCOE of ammonia SOFC technology compared to others such as hydrogen or methane?

Author Response

We thank the first reviewer for the fruitful comments and the insights for improving the paper. The quality of English was reviewed as suggested. The corrections are highlighted in the article version with reviews.

R1.1 The authors mentioned that ammonia could be produced using renewable energy sources. Please provide some references and methodologies in the paper.

Methodologies and references regarding the so-called “green ammonia” were reported in the paper.

R1.2 What are the potential environmental impacts of using ammonia in SOFC such as in mass production, storage tank and materials, toxicity, etc.?

In the aim of this study, the environmental impacts of the use of ammonia in SOFC were investigated only in power production process. The overall analysis of ammonia impacts requires a deeper and complete study at LCA level that are beyond the objective of this contribution. We consider such aspects extremely interesting and could be a significative follow up of the present work.

R1.3 Please include a graph showing the data and the correlation (Eq. (8))?

Data of Eq.8 are derived from literature. Ref. [32] reports the complete data and correlation between ammonia decomposition and temperature for the specific catalyst. The text was improved in the revised version of the manuscript.

R1.4 In Table 13, low temperature heat exchanger cost a lot more than the other heat exchanger? Any reasons?

I think the reviewer refers to Table 8. At lower temperature, the heat exchange is more complex and a bigger exchange area is required. This is the reason of the increase of the component cost. A comment was added in the text.

R1.5 What is the most economical value of LCOE of ammonia SOFC technology compared to others such as hydrogen or methane?

For the best of our knowledge, this is the first LCOE study dedicated to this kind of systems. The study compares the most economical calculated LCOE value (0.221 $ kWh-1) of NH3 based system with literature values of a methane based SOFC system (0.264 $ kWh-1).

Reviewer 2 Report

The operation of a SOFC based power system powered with ammonia is shown in this paper. In the paper experimental test and system design of SOFC was presented. The research supports the feasibility of ammonia fueled SOFC systems with reference to the carbon free energy market, specifically considering the potential development of green ammonia production.
The Authors presents the design and modeling of an ammonia-fed SOFC system based on a six cells SOFC stack. The Authors correlation with utilization of fuel, current density and ammonia decomposition at an operating temperature of 750°C was studied.
The study and the model developed in this study constitutes an important support for the design of an ammonia-fed SOFC energy system and provides indication for the sizing of the power unit.
After careful study of this document, it can be said that this is a very interesting material for readers.
This work is the next step to wider using of the FCs.
The research design is appropriate and the methods are adequately described.
References have been well selected the content of the paper, but it should be good to contain much more references.
Some suggestions follows:
It should be good to extend the references.
It should be good to rewrite the Conclusions chapter to for better understand this issue.

Author Response

We thank reviewer 2 for the comments. Introduction and conclusions were improved as suggested.

R2.1 The operation of a SOFC based power system powered with ammonia is shown in this paper. In the paper experimental test and system design of SOFC was presented. The research supports the feasibility of ammonia fueled SOFC systems with reference to the carbon free energy market, specifically considering the potential development of green ammonia production.
The Authors presents the design and modeling of an ammonia-fed SOFC system based on a six cells SOFC stack. The Authors correlation with utilization of fuel, current density and ammonia decomposition at an operating temperature of 750°C was studied.
The study and the model developed in this study constitutes an important support for the design of an ammonia-fed SOFC energy system and provides indication for the sizing of the power unit.
After careful study of this document, it can be said that this is a very interesting material for readers.
This work is the next step to wider using of the FCs.
The research design is appropriate and the methods are adequately described.

We thank the reviewer for the positive comments to our work.

R2.2 References have been well selected the content of the paper, but it should be good to contain much more references.

Number of references was increased as suggested by the reviewer.

Some suggestions follows:
R2.3 It should be good to extend the references.

References were extended.

R2.4 It should be good to rewrite the Conclusions chapter to for better understand this issue.

Conclusion were improved.

Reviewer 3 Report

Research objectives and focus are relevant for journal readership. Methodologies adopted and associated added value are clearly described. Minor English language revision (e.g., see introduction, line 121) is suggested. Finally, a schematic resuming the methodological workflow can be useful to improve readability and understanding.

Author Response

We thank the reviewer for the comments. English language was improved and the description of the methods was reviewed.

R3.1 Research objectives and focus are relevant for journal readership. Methodologies adopted and associated added value are clearly described. Minor English language revision (e.g., see introduction, line 121) is suggested.

We thank the reviewer for the positive comments. English quality was reviewed along the whole manuscript.

R3.2 Finally, a schematic resuming the methodological workflow can be useful to improve readability and understanding.

We thank the reviewer for the interesting suggestion. A schematic of the methodological workflow was added to the study.

Reviewer 4 Report

This paper considers the possible use of ammonia in fuel cell equipment to generate energy via hydrogen emitted after ammonia decomposition. It is estimated that a one-step process, i.e. decomposition and energy generation in SOFC stack is preferred to two-step one (with preliminary ammonia decomposition).

There are some discrepancies and non-correct statements in the text.

  1. The dimensions in Eqs (5,6) are inconsistent. Oranges and apples are compared. Power is measured in watts, but enthalpies in joules. What about heat losses (Q)?
  2. There is no reference for Eq.(8). The regression equation is empirical one, with no dimensions of the coefficients. What do they mean?
  3. How ADR approach is measured by degrees Celsius (Table 3)?
  4. There are no dimensions (%, possibly) in Tables4, 8, 9
  5. Careful edition of the English language is suggested.
  6. The abstract must be more concentrated and informative.

The paper could be published after answering the raised questions.

Author Response

We thank the reviewer for the comments. English language was improved and research design and methods description were reviewed.

R4.1 This paper considers the possible use of ammonia in fuel cell equipment to generate energy via hydrogen emitted after ammonia decomposition. It is estimated that a one-step process, i.e. decomposition and energy generation in SOFC stack is preferred to two-step one (with preliminary ammonia decomposition).

We thank the reviewer for the nice comments.

There are some discrepancies and non-correct statements in the text.R4.2 The dimensions in Eqs (5,6) are inconsistent. Oranges and apples are compared. Power is measured in watts, but enthalpies in joules. What about heat losses (Q)?

Equations description was improved and corrected as suggested by the reviewer.

R4.3 There is no reference for Eq.(8). The regression equation is empirical one, with no dimensions of the coefficients. What do they mean?

Reference of Eq. (8) is [26]. The regression is empirical obtained from the data published in literature. The results is dimensionless and TADR has to be introduced in °C. A comment was added in the text.

R4.4 How ADR approach is measured by degrees Celsius (Table 3)?

ADR approach is a temperature difference at the entrance of the ADR and is measured in °C.

R4.5 There are no dimensions (%, possibly) in Tables4, 8, 9

Value reported with no dimension units in Tables 4,8 and 9 such as Uf, Uox or BOR are ratio values dimensionless.

R4.6 Careful edition of the English language is suggested.

A deep review on English language was performed and the quality of English improved.

R4.7 he abstract must be more concentrated and informative.

Abstract was reviewed.